# DNA Metabarcoding as a Tool for Disentangling Food Webs in Agroecosystems

**DOI:** 10.3390/insects11050294

**Published:** 2020-05-11

**Authors:** Ahmadou Sow, Julien Haran, Laure Benoit, Maxime Galan, Thierry Brévault

**Affiliations:** 1Département de Biologie Animale, Faculté des Sciences et Techniques, Université Cheikh Anta Diop, Dakar, Senegal; 2CIRAD, UMR CBGP, F-34398 Montpellier, France; julien.haran@cirad.fr (J.H.); laure.benoit@cirad.fr (L.B.); 3CBGP, INRAE, CIRAD, IRD, Institut Agro, Université de Montpellier, F-34988 Montpellier, France; maxime.galan@inra.fr; 4INRA, UMR CBGP, F-34398 Montpellier, France; 5CIRAD, UPR AIDA, Centre de recherche ISRA-IRD, Dakar, Senegal; thierry.brevault@cirad.fr; 6AIDA, Univ Montpellier, CIRAD, F-34398 Montpellier, France

**Keywords:** conservation biological control, trophic network, intra-guild predation, arthropod diets, feces analysis, DNA sequencing, millet-based agroecosystem

## Abstract

Better knowledge of food webs and related ecological processes is fundamental to understanding the functional role of biodiversity in ecosystems. This is particularly true for pest regulation by natural enemies in agroecosystems. However, it is generally difficult to decipher the impact of predators, as they often leave no direct evidence of their activity. Metabarcoding via high-throughput sequencing (HTS) offers new opportunities for unraveling trophic linkages between generalist predators and their prey, and ultimately identifying key ecological drivers of natural pest regulation. Here, this approach proved effective in deciphering the diet composition of key predatory arthropods (nine species.; 27 prey taxa), insectivorous birds (one species, 13 prey taxa) and bats (one species; 103 prey taxa) sampled in a millet-based agroecosystem in Senegal. Such information makes it possible to identify the diet breadth and preferences of predators (e.g., mainly moths for bats), to design a qualitative trophic network, and to identify patterns of intraguild predation across arthropod predators, insectivorous vertebrates and parasitoids. Appropriateness and limitations of the proposed molecular-based approach for assessing the diet of crop pest predators and trophic linkages are discussed.

## 1. Introduction

Crop pests cause substantial economic losses to agricultural production and thus threaten the increase of crop productivity needed to achieve long-term food and nutrition security [1,2]. In addition, excessive reliance on chemicals to control crop pests is not sustainable, and innovations relying on ecologically-based approaches are urgently needed. Conserving functional biodiversity and related ecosystem services, especially by controlling pests using their natural enemies, offers new avenues to tackle challenges for the sustainable intensification of food production systems [3,4,5]. Predation of crop pests by generalist predators, including arthropods and vertebrates, is a major component of natural pest control [6]. A particularly important trait of most generalist predators is that they can colonize crops early in the season by first feeding on alternative prey [7,8]. However, the breadth of the “generalist” diet entails some drawbacks for pest control, such as intra-guild predation [6,9,10]. A tuned diagnosis of diet breadth in generalist predators, including predation of non-pest prey, is thus needed to better disentangle food webs (e.g., exploitation competition and apparent competition) and ultimately to identify key drivers of natural pest control in agroecosystems. However, the importance of generalist predators in the food web is generally difficult to assess, due to the ephemeral nature of individual predator–prey interactions [9,11]. The only conclusive evidence of predation results from direct observation of prey consumption, identification of prey residues within predators’ guts [9], and analyses of regurgitates [12] or feces [13]. Metabarcoding via high-throughput sequencing (HTS) offers new opportunities for deciphering trophic linkages between predators and their prey within food webs [14,15]. Compared to traditional, time-consuming methods, such as microscopic or serological analyses, the development of DNA metabarcoding allows the identification of prey species without prior knowledge of the predator’s prey range. In addition, metabarcoding can also be used to characterize a large number of species in a single PCR reaction, and to analyze several hundred samples simultaneously [16]. Such an approach is increasingly used to explore the functional diversity and structure of food webs in agroecosystems [15,17,18,19,20]. Like other molecular-based approaches, metabarcoding only gives qualitative results on the presence/absence of prey species in the gut or fecal samples [21]. However, this knowledge of the identity of prey consumed by predators of the same species in a given environment enables a “pragmatic and useful surrogate for truly quantitative information” [22].

In the present study, we used metabarcoding as a tool for identifying prey species from arthropods and fecal samples of insectivorous vertebrates involved in the natural regulation of insect pests in a millet-based agroecosystem in Senegal. The analysis of DNA sequences enabled us to reconstruct a trophic network in sampled millet fields, from the diagnosis of diet composition of main predatory arthropods and insectivorous vertebrates. Millet-based agroecosystems are particularly relevant for the study of complex processes of natural regulation of crop pests, because of the existence of a diverse set of natural enemy communities in these agroecosystems, including arthropod predators, parasitoids, and insectivorous vertebrates [23,24,25,26,27,28]. In addition, millet production relies on pest regulation by natural enemies, in the absence of any insecticide application by farmers. Appropriateness and limitations of the proposed molecular approach for assessing the diet of crop pest predators and reconstructing trophic networks in a given agroecosystem are discussed in the light of the results.

## 2. Materials and Methods 

### 2.1. Field Sampling

Arthropod predators and fecal samples of insectivorous vertebrates were collected during the 2018 growing season in two (2 km-distant) millet fields (or in the vicinity for birds and bats) located in the “peanut basin” in Senegal (Bambey, 14°43’0.79” N; 16°30’5.56” O). Ground-crawling arthropod predators (i.e., ants and carabid beetles) were collected daily for a two-week period in millet fields at the panicle stage, using dry pitfall traps (12-cm diameter, 14-cm depth). In all, 20 traps separated by at least 4 m were installed in each field along four transects, to avoid interference between traps. Plant-dwelling arthropods (e.g., earwigs, bugs and spiders) were collected during the same period, with a mouth or a backpack aspirator (Bioquip Products, Compton, CA, USA). In all, 256 arthropod specimens belonging to nine species were collected (Table 1). They were individually stored at −20 °C for at least 24 h, to avoid regurgitation (the most important source of cross-contamination [12,29]), then conserved in 90% ethanol for subsequent DNA analysis. A plastic tarpaulin was placed under two neem trees, *Azadirachta indica*, for the village weaver bird, *Ploceus cucullatus* (Passeriformes, Ploceidae), and four palm trees, *Borassus aethiopum*, for the Mauritian tomb bat, *Taphozous mauritianus* (Chiroptera, Emballonuridae). Fecal samples were collected five times (26 August, 4–11–16–25 September) during the millet reproductive stage. A total of 92 samples for birds and 80 for bats were individually collected using clean cotton buds and placed in 2-mL microtubes filled with silica-gel granules to absorb moisture and prevent the development of molds and DNA degradation.

### 2.2. DNA Extraction

DNA of the arthropod specimens was extracted using the EZ–10 96-well plate DNA Kit (Biobasic Inc., Toronto, Canada). Here, the whole arthropod bodies were used for DNA extraction because arthropod gut dissection is very time-consuming and increases contamination risk [29,30]. The external body parts were wiped with cotton wool previously soaked in 10% bleach before extraction. All extractions were carried out according to the manufacturer’s protocol and by non-destructive lysis, corresponding to individual incubation overnight in a mixture of 300 µL of animal cell lysis solution and 20 µL of Proteinase K at 55 °C in an oven, with stirring.

DNA of fecal samples was extracted using a NucleoSpin 96 Plant II kit (Macherey-Nagel), according to slight modifications recommended by Zarzoso-Lacoste et al. (2018) [31]. This kit provides the best compromise between throughput DNA isolation (up to 192 samples in parallel), price and sequencing success for predators and prey [14]. All fecal samples (*n* = 172 including 80 bird and 92 bat samples) were frozen at −80 °C and bead-beaten for 2 × 30 s at 30 Hz on a Tissue Lyser (Qiagen) using a 5-mm stainless steel bead, then extracted. 

### 2.3. PCR and Illumina Sequencing

A 133 bp fragment of cytochrome c oxidase I (COI mini-barcode, [32]) was amplified from arthropods and fecal samples using the primers and two-step PCR protocol previously described by Galan et al. (2018) [14], then sequenced on a MiSeq Illumina platform (Illumina, San Diego, CA, USA), using two independent runs for arthropods and vertebrate fecal samples. This short mini-barcode was shown to effectively discriminate arthropod species in millet-based agroecosystems [28]. It also offered the advantage of amplifying degraded DNA or rare DNA. In addition, Corse et al. (2019) [33] and Tournayre et al. (2019) [34] showed that primers used to amplify this mini-barcode were among the best over dozens of COI primer sets tested in silico. Several negative controls (*n* = 10) were included, according to recommendations made by Galan et al. (2016) [35], such as: (i) negative controls for DNA extraction (NCext), (ii) negative controls for PCR (NCpcr), (iii) negative controls for indexing (NCindex: unused dual-index combinations), (iv) positive controls for PCR (PCpcr) and (v) positive controls for indexing (PCalien: DNA from beluga whale—*Delphinapterus leucas*—used to estimate the read misassignment frequency). We performed three technical replicates on each DNA extract of fecal and arthropod samples in order to control PCR stochasticity for rare DNA targets and to validate positive results [36].

A first amplification step of the short 133 bp COI fragment was carried out using two universal primers. For arthropods, primers (MG-LCO1490 5′-ATTCHACDAAYCAYAARGAYATYGG-3′ and MG-univ-R 5′- ACTATAAAARAAAYTATDAYAAADGCRTG-3′) adapted by Galan et al. (2018) [14] from the original primer sets designed by Gillet et al. (2015) [32] were used. For vertebrate feces, primers (MG2-LCO1490 5′-TCHAC**H**AAYCAYAARGAYATYGG-3′ and MG2-univ-R 5′-AC**Y**AT**R**AARA**ARATY**ATDAY**R**AADGCRTG-3′) adapted by Tournayre et al. (2019) [34], were used (the bases in bold indicate the modifications to improve the primers). PCR reactions were carried out in a final volume of 10 µL containing 5 µL of Multiplex Master Mix (Qiagen, Hilden, Germany), 0.5 μM of each primer and 2 µL of DNA. PCR conditions in this first step consisted of initial denaturation at 95 °C for 15 min followed by 40 cycles of denaturation at 94 °C for 30 s, hybridization at 45 °C for 45 s and extension at 72 °C for 30 s, and a final extension step at 72 °C for 10 min. A second PCR step was performed to add individual-specific multiplexing tags (called index i5 and index i7), consisting of short 8-bp sample-specific sequences and the Illumina adapters (called P5 and P7) at the 5’ ends of each amplified DNA fragment to the first PCR. As all the PCR products were mixed together (multiplexing) for MiSeq sequencing, indexes made it possible to identify the origin of sequences and reassign them to each sample (demultiplexing). This second PCR was carried out in a total volume of 10 µL containing 5 µL of Multiplex kit (Qiagen, Hilden, Germany), 0.7 μM of each primer, and 2 µL of products from the first PCR for each sample. PCR conditions consisted of an initial denaturation step at 95 °C for 15 min followed by 8 cycles of denaturation at 95 °C for 40 s, hybridization at 55 °C for 45 s and extension at 72 °C for 60 s, and a final extension step at 72 °C for 10 min.

The volume-to-volume mix of all the PCR products (4 µL per sample) was screened based on fragment length resulting from excision on 1.5% agarose gel. The pool of specific PCR products was cut to the expected size (328 bp corresponding to the size of the amplicon, including the gene-specific primers, sequencing primers, indexes and adaptors) under UV light. The resulting library was purified with the NucleoSpin Gel & PCR Clean-Up purification kit (Macherey-Nagel, Düren, Germany), quantified by quantitative PCR (KAPA kit, Kapa Biosystems, Waltham, MA, USA), then sequenced on a MiSeq sequencer with a 500v2 kit (Illumina, San Diego, CA, USA).

### 2.4. Sequence Analysis

Raw data from the Illumina MiSeq were deposited in the Zenodo data repository [37]. The paired-end sequencing data from the Illumina MiSeq system were processed using the method described by Sow et al. (2019) [27]. We processed the paired-end sequencing data from the Illumina MiSeq system with the FROGS pipeline [38]. We preliminarily used a home-made script available at Dryad data repository [39] to merge pair sequences into contigs with FLASH v. 1.2.11 [40], and trim primers with cutadapt v. 1.9.1 [41]. This script is particularly useful when using FROGS on data produced with primers that vary in size between reads, which is not covered by the pre-process step of FROGS. In our study, size variation in primers resulted from the addition of heterogeneity spacers in adapters during the library construction to shift the reading frame [42]. In FROGS, we then filtered sequences by length (expected value of 133b ± 10b), dereplicated sequences, removed chimeras using the algorithm of Edgar et al. (2011) [43] implemented in VSEARCH v. 1.1.3 and clustered sequences with SWARM v. 1.3.0 using a local clustering threshold using the default d-value (d = 1) [44].

Taxonomic affiliations for each OTU (Operational Taxonomic Unit) were returned using a previously constructed reference database [28]. The database consisted of 234 sequences of 658 bp of the mitochondrial cytochrome c oxidase 1 from species involved in the trophic web of the millet head miner (MHM, key millet pest) in Senegal [28]. Sequences absent from our database were compared by NCBI Blast+ on public databases (BOLD Systems v. 3 [45], and GenBank [46]). The identification was considered as ‘valid’ from a similarity threshold of 97 to 100%. Below that threshold, the OTUs were considered unidentified. Filtering for false positives was carried out as proposed by Galan et al. (2016) [35]. In short, we discarded positive results associated with sequence counts below two OTU-specific thresholds, which checked respectively for cross-contamination between samples (using, respectively, twelve and six negative controls for the arthropod samples and vertebrate fecal samples) and incorrect assignment due to the generation of mixed clusters on the flowcell during Illumina sequencing, using a false index-pairing rate for each PCR product of 0.02%, based on estimates from Galan et al. (2016) [35] (see Appendix A). For each sample, OTUs found in at least two of the three PCR replicates were considered positive, and OTUs found in only one of the three PCR replicates were removed. This strategy is the best compromise between eliminating false positives and conserving low-biomass prey [47]. Lastly, for each identified prey, the number of sequences obtained for each PCR replicate from a sample were summed.

## 3. Results

### 3.1. Sequence Analyses

We conducted Illumina MiSeq run to sequence 256 arthropod samples. In all, 4,333,970 reads were generated (mean = 16,929.57, SD = 3264.68) and, after sequence filtering, 4,274,516 reads (mean = 16,697.35; SD = 3262.51) remained. In all, 516 OTUs corresponding to environmental contaminations, chimera, pseudogenes and sequences for which no relevant match was obtained were also excluded from the analysis. Here, sequences of one pest, the millet head miner, *Heliocheilus albipunctella* (Lepidoptera, Noctuidae), were deleted after filtering due to critical contamination of part of the negative controls (NCext and NCpcr) for this taxon. On the 181 remaining OTUs, 91 OTUs were identified to species or genus level using our own reference database and public reference databases (Appendix A). We then conducted another Illumina MiSeq run to sequence 172 vertebrate fecal samples (92 from bats, 80 from birds). In all, 2,085,699 reads were generated (mean = 11,630.30; SD = 5744.10) and, after sequence filtering, 1,957,455 reads (mean = 11,380.55; SD = 5240. 57) remained. In all, 894 unidentified OTUs and 28 OTUs corresponding to environmental contaminations, such as sequences of rodents, lizards (*Gama gama*), and humans, were also excluded from the analysis. Of the 522 remaining OTUs, 154 OTUs were identified to species or genus level (Appendix A). The raw and filtered tables of abundance for these two sequencing runs are available in the Zenodo data repository (see link in Appendix A).

### 3.2. Diet Composition of Predatory Arthropods

After assigning the identified OTUs as species, 27 arthropod prey taxa were identified from the nine arthropod predators (*n* = 256) sampled from millet fields (Table 1). The diet of predatory arthropods was assessed only on 25.4% (range 4.0–58.3) of the total number of samples tested. Among positive samples, most reads belonged to predatory arthropods themselves (97.8% of the reads obtained) and their diets were described only on 2.2% (range 0.1–7.0) of total reads obtained (Figure 1A). The mean number of prey taxa detected per sample was the highest in carabid beetles (1.1, range 0–4), ants (0.5, range 0–3) and spiders (0.4, range 0–2) and the lowest in the remaining predators including anthocorid bugs (0.2, range 0–2), pentatomid bugs (0.2, range 0–2), and earwigs (0.04, range 0–1) (Figure 1B). Across predatory arthropods, a high diversity of arthropod preys was observed in spiders (14 species belonging to 11 families and seven orders), carabid beetles (nine species belonging to eight families and six orders), ants (seven species belonging to six families and four orders), and anthocorid bugs (six taxa belonging to five families and two orders). In contrast, the diversity of prey species identified in earwigs (three species belonging to three families and three orders) and pentatomid bugs (three species belonging to three families and two orders) was relatively low. Lepidoptera (detected on six of nine predators tested), Hemiptera (detected on five of nine predators tested), Diptera (detected on four of nine predators tested) and Coleoptera (detected on four of nine predators tested) were the most common insect prey taxa detected from predatory arthropods (Figure 2).

### 3.3. Diet Composition of Insectivorous Vertebrates

In all, 110 insect prey taxa were identified from fecal samples of insectivorous birds and bats (Table 2). Diet of birds and bats was assessed on 13.8 (11 out of 80) and 93.0% (86 out of 92) of samples tested, respectively. Few reads belonged to insect prey (10.5%) in bird fecal samples, whereas most reads belonged to insect prey (61.0% of the reads obtained) in bat fecal samples (Figure 1A). A lower diversity of prey species was observed in fecal samples of birds (0.4, range 0–4) compared to that of bats (5.0, range 0–16) (Figure 1B).

In bird fecal samples, 13 species distributed in six orders were identified (Table 2, Figure 3A). Most prey species belonged to Lepidoptera (six species belonging to three families) and Hymenoptera (three species belonging to three families), but Coleoptera (one species), Dermaptera (one species), Diptera (one species) and Hemiptera (one species) were also identified. Earwigs (*Forficula senegalensis*) were the most frequent prey species detected from bird fecal samples (6.3% of positives samples). Among herbivorous species, most were crop pests (e.g., *H. albipunctella*). In bat fecal samples, most prey species were Lepidoptera (81 species belonging to 15 families), but Orthoptera (10 species belonging to two families), Hemiptera (four species belonging to two families), Neuroptera (three species belonging to one family), Blattodae (two species belonging to one family), Coleoptera (one species), Diptera (one species), and Hymenoptera (one species) were also detected (Table 2, Figure 3A). Among Lepidopterans, species mainly belonged to Noctuidae (32.1% of prey taxa) followed by Erebidae (22.2%), Crambidae (9.9%) and Geometridae (8.8%). Two Crambidae species belonging to the genus Achyra (*A. coelatalis* and *A. nudalis*) were the most frequently observed, with 51% of positive samples. Among phytophagous species for which host plants have been documented, 63% were crop pests. For both insectivorous vertebrates, composition of prey species differed according to sampling date (Figure 3B).

### 3.4. Qualitative Trophic Network and Interactions

Qualitative trophic networks were designed based on diet composition of arthropod and vertebrate predators. Two main patterns of intraguild predation were identified: arthropod predators vs. arthropod predators, and insectivorous vertebrates vs. arthropod predators (Figure 4).

## 4. Discussion

In food web ecology, “who eats whom” is a fundamental issue for gaining a better understanding of the complex trophic interactions existing between pests and their natural enemies within a given ecosystem [11,48]. Results here provided bring essential data for characterizing the functional diversity and structure of the food web associated with a couple of millet fields in Senegal. The dietary analysis of arthropod and vertebrate predators enables the identification of key predators involved in the natural control of arthropod pests and gives insights into the breadth of their diet (generalist vs. specialist) and intraguild predation.

### 4.1. Taxonomic Affiliation and Diet Analysis

The reliability of biological information provided by metabarcoding mostly depends on the non-arbitrary measures taken to limit PCR bias, which may lead to misinterpretations. In this study, non-arbitrary filters for false positive results were applied as recommended by Galan et al. (2018) [14]. A proportion of 65% of the OTUs obtained after filtering could not be identified using either conventional databases such as BOLD and Genbank or our reference database [28]. This limitation generated false negative results, as it was not possible to detect the occurrence of species not included in the reference databases, mostly because African entomofauna is still poorly documented. The trophic network obtained was thus constructed from 35% of the identified OTUs. However, some unidentified OTUs might be pseudogenes or non-target sequences that would not be referenced within an exhaustive database. The reconstitution of food webs by metabarcoding depends on the ability to identify the majority of sequences of interest, which in turn mainly depends on the completeness of the reference databases and on the resolution of the minibarcode used (i.e., using several genes, see [15,47,49]). In this study, 133 bp of COI was used as a very resolutive fragment for most arthropods [28,32] and vertebrates [14].

Technical replications, which showed a high proportion of stochastic positive results, are also necessary to limit PCR bias in metabarcoding, increase the detection of low biomass taxa, and thus reduce subsequent underestimation of predation [47]. The instability with PCR replicates was mainly observed in arthropod predators. This was probably related to the presence of a large quantity of DNA of the analyzed predator itself, which decreased the detectability of prey. On the contrary, prey DNA was present in large quantities in the fecal samples of insectivorous vertebrates. These differences of DNA amplification between arthropod and vertebrate predators may also result from PCR bias due to primers [33]. These observations highlight the importance of considering enough specimens and including biological replicates. At last, it is important to note that DNA-based techniques cannot detect cannibalism, which may be substantial in some predators [50]. 

In this study, DNA extraction from the whole body of predatory arthropods could lead to possible contamination by the exoskeleton (i.e., DNA outside the body) of these arthropods. Nevertheless, precautions were taken to avoid potential external contamination of arthropod samples by systematically individualizing samples during field sampling and avoiding regurgitation by placing “living” samples at −20 °C before storage in alcohol [51]. Critical contamination of the sequence of the millet head miner (MHM, millet pest) during the MiSeq sequencing run on predatory arthropods is another limitation of this study. However, generalist predators of the MHM have been extensively documented [23,24,28,52]. Here, the main objective was to identify alternative prey of predatory arthropods, diet composition of insectivorous vertebrates, and extent of intra-guild predation in the studied agroecosystem.

### 4.2. Diet Composition of Predatory Arthropods

The “generalist” diet was confirmed for all tested predatory arthropods. However, the degree of polyphagy differed across species. More diversity of prey taxa was observed from “true predators” including spiders, carabid beetles, anthocorid bugs, and ants (except *Pachycondula* sp., likely due to the low number of samples). These predators preyed on a wide array of arthropod orders (27 prey taxa belonging to 18 families and 9 orders were identified). Spiders [53,54,55], carabid beetles [17,56,57,58], anthocorid bugs [59], and ants [60] are considered as effective biological control agents in several crops. Their diet composition showed the presence of crop pests (e.g., Lepidoptera, Hemiptera, and Thysanoptera), but also non-pest arthropods (e.g., wasps, beetles). The anthocorid predatory bug, *Orius maxidentex*, preyed on four Lepidopteran prey taxa, including *Ephestia kuehniella* and *Masalia nubila,* which are secondary pests of pearl millet [23,61]. Carabid beetles preyed also on two important pests of sorghum and maize, namely the common blossom thrips (*Frankliniella schultzei,* Thripidae) and shoot flies (*Atherigona sp*., Muscidae). The diet composition of earwigs, *F. senegalensis* (Forficulidae), and pentatomid bugs, *C. curtana* (Pentatomidae), supported their facultative predator status. However, the beneficial status of these omnivorous insects, often secondary pests, has often been controversial [62]. Populations of earwigs mainly feed on millet pollen [52] to which they add insects (here, one Lepidopteran species and two mirid bugs) during their breeding period. Due to its high abundance in millet fields and the non-limitative quantity of pollen for millet pollination, this species should be considered as beneficial for pearl millet. This is probably also the case for *C. curtana* which generally feeds on millet grains but can also prey on small arthropods. Like ‘true predators’ that can persist on crops in the absence or low density of target pests by feeding on alternative prey, these omnivore arthropods (namely ‘false predators’) can persist in the absence of prey by feeding on plants, enabling them to maintain high populations before pest arrival.

### 4.3. Diet Composition of Insectivorous Vertebrates

The analysis of DNA from fecal samples of the Mauritian tomb bat revealed a diet mainly composed of moths (79% of total species detected), as shown by Kingdon (1974) [63], of which two species of the genus *Achyra* were dominant (81 out of 103 prey taxa). The majority of Lepidopteran pests identified are among the most economically important crop pests in Senegal, including armyworms (*Spodoptera* spp., see [64,65,66]) and earworms (*Helicoverpa armigera*, see [67]). As expected, DNA of the main pest of pearl millet, the millet head miner, was detected (8% of positive samples). The Mauritian tomb bat contributes to the regulation of many nocturnal millet pests because, unlike arthropod predators and birds, insectivorous bats can feed on a large quantity of individuals (up to 16 prey taxa detected in a single sample), thus limiting egg-laying on millet panicles. In addition, the activity of insectivorous bats has been shown to be strongly correlated with the abundance of arthropods in agricultural systems [68]. The species is characterized by very effective hunting aptitudes, including excellent vision, nocturnal hunting, echolocation of prey, and fast flight [69,70], which make it an excellent predator of noctuid moths. Our field observations revealed that this generalist predator was able to hunt under public lights in villages, in addition to its ability to hunt in total darkness or to perform hovering flights to catch insects on millet panicles [26]. The diet analysis of bats revealed a seasonal fluctuation of prey diversity (richness), with a particular increase at the early reproductive stage of millet (heading).

The diet of village weaver birds was also composed of a diversity of arthropods. Of the 13 arthropod species identified, Lepidopterans were the most common species (6 out of 13 prey taxa), followed by earwigs (23.3% of positive samples). This is consistent with observations made by Bruggers et al. (1985) [71], who reported that adults feed mainly on insects to which they add seeds of wild plants. Unlike Mauritian tomb bats, which are solitary and strictly insectivorous, village weavers are omnivorous and gregarious [72]. Seasonal fluctuation of prey composition was also observed in village weaver birds, with a particular shift from Dermaptera to Lepidoptera during the reproductive stage of millet. Comprehensive analysis of diet composition in time should include chicks that feed exclusively on insects [71]. Colonies of village weavers settled a few weeks in the study area for breeding, from emergence to maturity of millet panicles [26]. More research is needed to better understand the migratory flow of these insectivorous vertebrates in and beyond Senegal, and its importance in terms of ecosystem services such as pest control.

### 4.4. Qualitative Trophic Network and Interactions

Our approach also provided information on interactions between natural enemies, particularly asymmetric intraguild predation. Diet analyses revealed predation across generalist arthropod predators [e.g., spiders (sp.*1*) and ants (*Pachycondula* sp.) feeding on carabids (*B. scalaris*) and predation of carabids on spiders belonging to the family Gnaphosidae)] or between parasitoids and arthropod predators (e.g., *C. curtana* parasitized by *Cylindromyia bicolor*). Our results showed that the Mauritian tomb bat fed on three Neuropteran species, and the village weavers frequently fed on earwigs and ants. Such intra-guild predation could affect the overall abundance of natural enemies and thus biological control of phytophagous arthropods [73,74]. However, according to a meta-analysis conducted by Mooney et al. (2010) [75], this relation does not necessarily extend to insectivorous vertebrates. Authors showed that effects of insectivorous vertebrates on predatory and phytophagous arthropods were positively correlated and that, although insectivorous vertebrates fed as intraguild predators, strongly reducing arthropod predators, they nevertheless suppressed herbivores and indirectly reduced plant damage. Such effects were strongest on arthropods and plants in communities with abundant arthropod predators and strong intraguild predation. 

The parasitoid, *Pristomerus pallidus* and the hyerparasitoid, *Perilampus* sp. (see [28]), were also detected in the gut content of birds. It is, however, probable that those birds fed on already parasitized Lepidoptera larvae. This illustrates the ability of metabarcoding to identify unexpected interactions between natural enemies.

## 5. Conclusions

Better knowledge of food webs and related ecological processes is fundamental for understanding the functional role of biodiversity in ecosystems [76]. This is particularly true in agroecosystems to promote environment-friendly models for crop protection inspired by natural pest control [77]. However, it is generally difficult to decipher the impact of predators in food webs, as they often leave no direct evidence of their activity. In this study, a metabarcoding approach enabled us to reconstruct qualitative trophic networks from field sampling of generalist predators. This approach proved effective in detecting preys in arthropods and feces of birds and bats. Key information on the diet breadth and preference of predators, trophic linkages within food web, including intra-guild predation, and potential apparent competition, were also provided. Such a qualitative approach should, however, be supplemented with quantitative assessments, such as spatio-temporal abundance or predation rates [25] to identify predators contributing to the suppression of the pest population and to validate the relevance of such an ecosystem service for sustainable pest management. There is also a need to improve our knowledge on the life system of key natural enemies (e.g., resource requirements and dispersal), so that specific conservation measures, such as habitat management, can be designed to promote their impact as part of biological control strategies.

## Figures and Tables

**Figure 1 insects-11-00294-f001:**
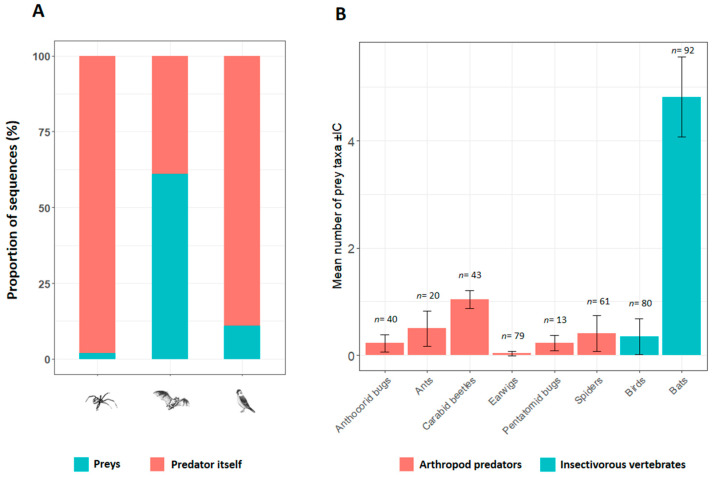
(**A**) Proportion of sequences attributed to predators themselves (arthropods, bats and birds) or its preys. (**B**) Mean number of prey taxa detected in arthropod guts or fecal samples of insectivorous vertebrates. IC: 95% confidence interval. *n*: number of samples.

**Figure 2 insects-11-00294-f002:**
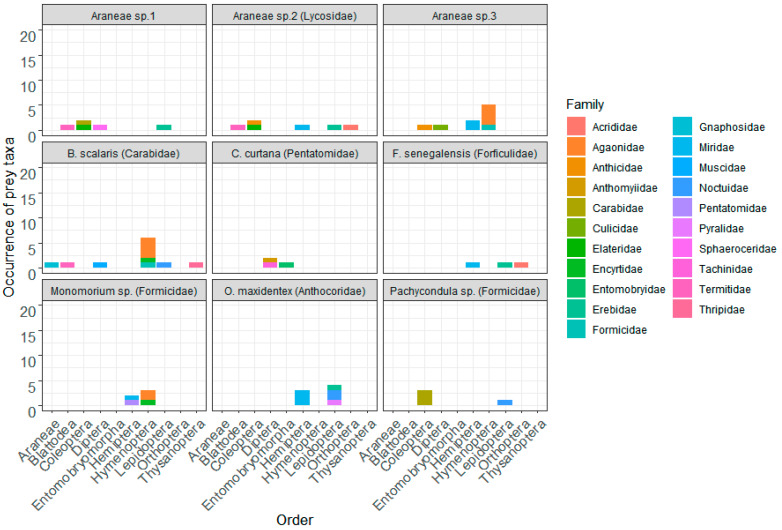
Diet composition of predatory arthropods inferred from DNA sequences (identified at genus or species level) detected. Occurrence of prey taxa is the cumulated number of OTUs identified.

**Figure 3 insects-11-00294-f003:**
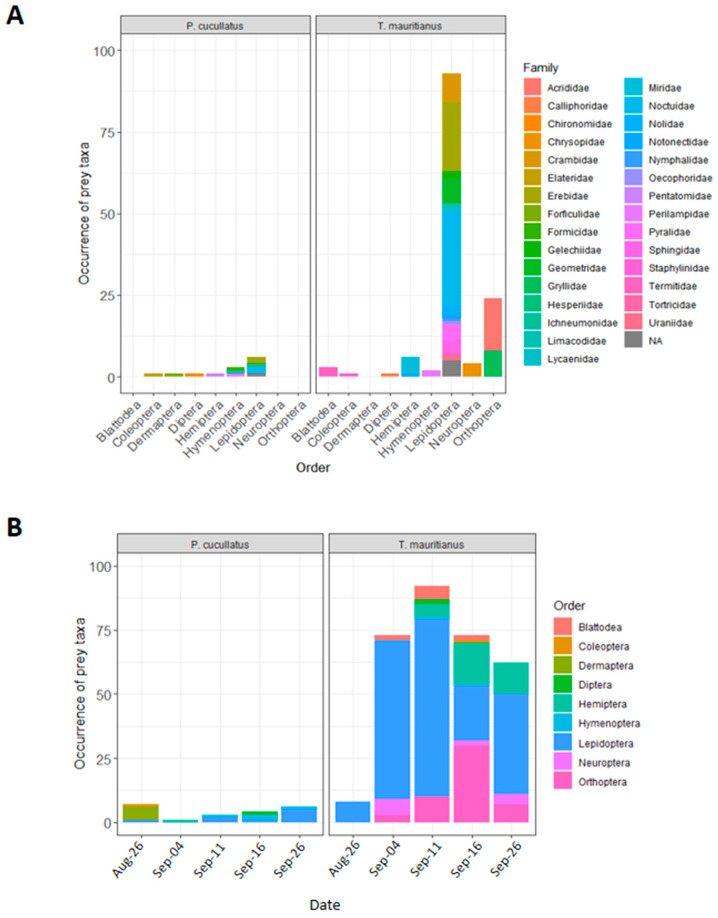
(**A**) Diet composition of insectivorous birds (*Ploceus cucullatus*) and bats (*Taphozous mauritianus*) inferred from DNA sequences detected in fecal samples. (**B**) Diet composition as a function of sampling date throughout the reproductive stage of millet. Occurrence of prey taxa is the cumulated number of OTUS identified from fecal samples.

**Figure 4 insects-11-00294-f004:**
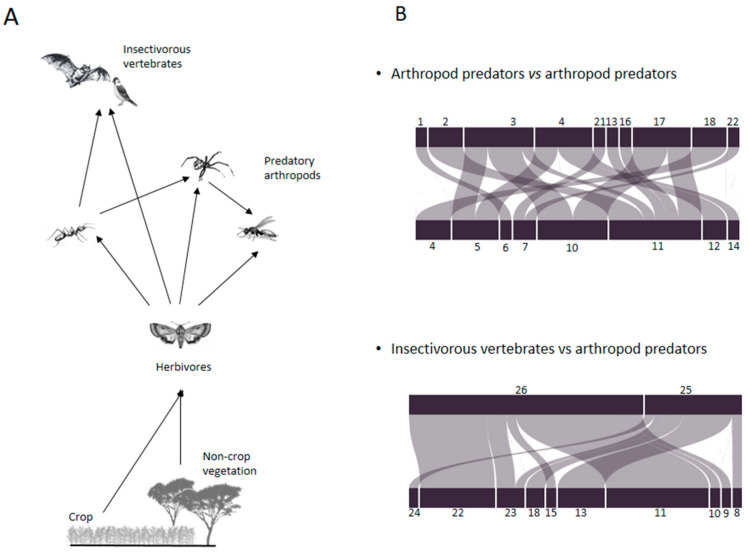
(**A**) Qualitative trophic networks of arthropod and vertebrate predators collected from a millet-based agroecosystem in Senegal. Arrows represent biomass flow between predators and preys. (**B**) Intraguild interactions. Arthropod predators: 1: *Araneae* sp.1. 2: *Araneae* sp.2. 3: *Araneae* sp.3. 4: *Bradybeanus scalaris*. 5: *Camponotus* sp. 6: *Carabidae* sp. 7: *Carbula curtana*. 8: *Carbula* sp. 9: *Chrysoperla* sp. 10: *Creontiades pacificus*. 11. *Creontiades pallidus*. 12: *Encyrtidae* sp. 13: *Forficula senegalensis*. 14: *Gnaphosidae* sp. 15: *Mallada signatus*. 16: *Monomorium* sp. 17. *Orius maxidentex*. 18: *Pachycondyla* sp. 19: *Philonthus discoideus*. 20: *Plesiochrysa atalotis*. Parasitoids of arthropods: 21: *Cylindromyia bicolor*. 22: *Pegoplata nigroscutellata*. 23: *Perilampus* sp. 24: *Pristomerus pallidus*. Insectivorous vertebrates: 25: *Ploceus cucullatus*. 26: *Taphozous mauritianus*.

**Table 1 insects-11-00294-t001:** Taxa and number of positive samples of prey species detected from nine arthropod predator taxa (*n* = 256) collected from millet fields in Senegal.

Predator	Number of Samples Analyzed	Prey			
Order (Family) *Species*		Order	*Species*	Blast Identity (%)	Positive Samples (%)
**Araneae (unidentified)**	12	Blattodea	*Odontotermes* sp.	100	8.3
*Sp.1*		Coleoptera	*Carabidae* sp.	99	8.3
			*Elateridae* sp.	99	25.0
		Diptera	*Rachispoda* sp.	99	8.3
		Lepidoptera	*Hypena masurialis*	100	8.3
*Sp.2*	30	Blattodea	*Odontotermes* sp.	100	10.0
		Coleoptera	*Elateridae* sp.	98	3.3
			*Omonadus floralis*	100	3.3
		Hemiptera	*Creontiades pallidus*	99	3.3
		Lepidoptera	*Hypena masurialis*	100	3.3
		Orthoptera	*Calliptamus barbarus*	100	3.3
*Sp.3*	19	Coleoptera	*Omonadus formicarius*	99	5.3
		Diptera	*Culex* sp.	100	5.3
		Hemiptera	*Creontiades pacificus*	100	5.3
			*Creontiades pallidus*	99	10.5
		Hymenoptera	*Apocryptophagus testaceus*	98	5.3
			*Camponotus* sp.	100	10.5
			*Ceratosolen fusciceps*	99	15.8
**Coleoptera (Carabidae)**	43	Araneae	*Gnaphosidae* sp.	100	2.3
*Bradybeanus scalaris*		Blattodea	*Odontotermes* sp.	100	9.3
		Diptera	*Atherigona* sp.	97	9.3
		Hymenoptera	*Camponotus* sp.	100	4.7
			*Ceratosolen fusciceps*	99	14.0
			*Encyrtidae* sp.	100	4.7
			*Sycophaga testacea*	99	4.7
		Lepidoptera	*Aegoceropsis* sp.	100	2.3
		Thysanoptera	*Frankliniella schultzei*	100	7.0
**Dermaptera (Forficulidae)**	79	Hemiptera	*Creontiades pallidus*	100	1.3
*Forficula senegalensis*		Lepidoptera	*Hypena masurialis*	100	1.3
		Orthoptera	*Calliptamus barbarus*	100	1.3
**Hemiptera (Pentatomidae)**	13	Diptera	*Cylindromyia bicolor*	100	7.7
*Carbula curtana*			*Pegoplata nigroscutellata*	100	7.7
		Entomobryomorpha	*Entomobrya ligata*	100	7.7
**Hemiptera (Anthocoridae)** *Orius maxidentex*	40	Hemiptera	*Creontiades pacificus*	100	5.0
			*Creontiades pallidus*	99	7.5
		Lepidoptera	*Amyna axis*	100	2.5
			*Ephestia kuehniella*	100	2.5
			*Hypena masurialis*	100	2.5
			*Masalia nubila*	97	2.5
**Hymenoptera (Formicidae)**	15	Hemiptera	*Carbula* sp.	97	6.7
*Monomorium* sp.			*Creontiades pallidus*	99	6.7
		Hymenoptera	*Apocryptophagus testaceus*	98	6.7
			*Ceratosolen fusciceps*	99	6.7
			*Cryptanusia* sp.	98	13.3
*Pachycondyla* sp.	5	Coleoptera	*Bradybaenus scalaris*	100	60.0
		Lepidoptera	*Aegoceropsis* sp.	100	20.0


Prey detected from arthropod predators with a > 97% threshold of correspondence with our own reference database (Sow et al., 2018), BOLD or GenBank. All predator species were identified by specialists, except spiders (in progress).

**Table 2 insects-11-00294-t002:** Taxa and number of positive samples of prey species detected in feces of two major insectivorous vertebrates collected in the surrounding environment of millet fields in Senegal.

Predator	Number of Samples	Prey			
Order (Family) *Species*		Orders Family	Species	Blast Identity (%)	Postive Samples (%)
**Chiroptera (Emballonuridae)**	92	Blattodea	*Macrotermes subhyalinus*	100	2.17
*Taphozous mauritianus*			*Odontotermes* sp.	100	2.72
		Coleoptera	*Philonthus discoideus*	98	1.09
		Diptera	*Chrysomya marginalis*	99	3.26
		Hemiptera	*Anisops sardeus*	100	2.17
			*Campylomma* sp.	100	16.30
			*Creontiades pacificus*	100	1.09
			*Creontiades pallidus*	99	3.99
		Hymenoptera	*Perilampus* sp.	100	1.09
		Lepidoptera	*Achyra coelatalis*	99	38.04
			*Achyra nudalis*	100	13.04
			*Acontia gratiosa*	100	3.26
			*Acontia* sp.	100	3.26
			*Adisura bella*	100	11.41
			*Adoxophyes thoracica*	99	3.26
			*Agrius convolvuli*	100	3.26
			*Amyna axis*	100	9.78
			*Ancylosis nubeculella*	100	7.61
			*Anomis flava*	100	3.80
			*Antheua simplex*	99	3.26
			*Asota heliconia*	98	1.63
			*Bastilla angularis*	100	1.09
			*Brachynemata* sp.	99	1.09
			*Cadra* sp.	100	1.09
			*Caryocolum petrophila*	99	1.09
			*Cerastis rubricosa*	98	2.17
			*Chrysodeixis acuta*	100	1.09
			*Condica capensis*	100	3.26
			*Coniesta ignefusalis*	100	1.09
			*Daphnis nerii*	100	1.09
			*Deltote* sp.	98	1.09
			*Duponchelia fovealis*	100	2.17
			*Ebertidia haderonides*	98	1.09
			*Ectopatria* sp.	100	1.09
			*Ericeia inangulata*	99	2.17
			*Franclemontia interrogans*	99	1.09
			*Garella nilotica*	100	5.43
			*Grammodes stolida*	100	13.04
			*Helicoverpa armigera*	99	3.26
			*Helicoverpa assulta*	100	15.22
			*Heliocheilus albipunctella*	100	7.61
			*Hippotion gracilis*	100	2.17
			*Hyles* sp.	100	3.26
			*Hypena holophaea*	100	1.09
			*Hypena laceratalis*	100	4.35
			*Hypena masurialis*	100	4.35
			*Isturgia disputaria*	100	3.26
			*Isturgia pulinda*	100	4.35
			*Lamoria imbella*	100	3.26
			*Lepidoptera*	100	1.09
			*Leucania commoides*	100	1.09
			*Leucania* sp.	98	1.09
			*Lophocrama phoennicochlora*	100	1.09
			*Marasmia poeyalis*	100	4.35
			*Maurilia* sp.	98	1.09
			*Morosaphycita oculiferella*	98	2.17
			*Mythimna* sp.	100	7.61
			*Noctuidae*	100	7.61
			*Ortaliella palaestinensis*	97	1.09
			*Pasiphila derasata*	100	1.09
			*Pericyma* sp.	100	1.09
			*Phazaca theclata*	100	2.17
			*Polydesma umbricola*	100	1.09
			*Polypogon fractalis*	100	1.09
			*Psilopleura vittata*	99	1.09
			*Pyrausta* sp.	100	1.09
			*Rhabdophera exarata*	99	2.17
			*Rhesala moestalis*	100	2.17
			*Scoliopteryginae sp*	100	2.17
			*Scopelodes sericea*	98	1.09
			*Scopula adelpharia*	100	1.63
			*Sesamia calamistis*	100	8.70
			*Sphingomorpha chlorea*	100	1.09
			*Spodoptera cilium*	100	4.35
			*Spodoptera exigua*	100	7.61
			*Spodoptera littoralis*	100	1.09
			*Spodoptera* sp.	100	2.17
			*Spoladea recurvalis*	100	10.87
			*Stegasta* sp.	98	1.09
			*Sympis rufibasis*	99	1.09
			*Tegostoma* sp.	100	8.70
			*Theclinae*	97	1.09
			*Thylacoptila paurosema*	100	4.35
			*Thysanopyga cermala*	99	1.09
			*Traminda neptunaria*	100	1.09
			*Trichoplusia ni*	100	10.87
			*Trigonodes hyppasia*	100	3.26
			*Trisuloides* sp.	99	1.09
			*Uraba lugens*	98	2.17
			*Vanessa cardui*	100	1.09
		Neuroptera	*Chrysoperla sp*	98	1.09
			*Mallada signatus*	98	4.35
			*Plesiochrysa atalotis*	99	1.09
		Orthoptera	*Acheta domesticus*	99	2.17
			*Arcotylus longipes*	99	1.09
			*Gryllus bimaculatus*	100	5.43
			*Gryllus campestris*	98	1.09
			*Locusta migratoria*	100	1.09
			*Oecanthus pellucens*	99	1.09
			*Oedaleus decorus*	99	3.73
			*Oedaleus* sp.	98	1.09
			*Stenohippus maculifemur*	99	1.09
			*Trilophidia conturbata*	99	1.45
**Passeriformes (Ploceidae)**	80	Coleoptera	*Elateridae*	99	1.25
*Ploceus cucullatus*		Dermaptera	*Forficula senegalensis*	100	6.25
		Diptera	*Bryophaenocladius* sp.	100	1.25
		Hemiptera	*Carbula* sp.	98	1.25
		Hymenoptera	*Pachycondyla* sp.	100	2.5
			*Perilampus* sp.	100	1.25
			*Pristomerus pallidus*	100	1.25
		Lepidoptera	*Heliocheilus albipunctella*	100	2.5
			*Lepidoptera*	98	1.25
			*Pelopidas mathias*	100	2.5
			*Scoliopteryginae sp*	100	1.25
			*Sphingomorpha chlorea*	100	2.5
			*Spodoptera cilium*	100	1.25

Prey detected in fecal samples of insectivorous vertebrates with a >97% threshold of correspondence with our own reference database [28], BOLD or Genbank.

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
