# Peer review of "DNA Metabarcoding as a Tool for Disentangling Food Webs in Agroecosystems"

_insects, 2020, doi:10.3390/insects11050294_

Round 1

Reviewer 1 Report

The idea presented in this work is very fascinating and the scientific interest would be great. I also appreciated reading the introduction and discussion: both well-written and easily understandable.

Unfortunately, there are two major issues that undermine the whole work.

Firstly, while the authors keep referring to the "gut content" of the insects, there is no scientific proof they indeed extracted DNA from the guts. DNA extraction was performed using a non-destructive DNA extraction, without separating the gut from the rest of the body AND without cleaning/sterilizing the surface of the insects. In turn, this means the DNA extracted could be entirely from the surface of the insects and, therefore, the result of contaminations. Why didn't the authors try to dissect the insects and extract DNA just from the gut?

Supporting the idea of contamination as opposed to gut content is, for example, the fact that some of the results present Lepidoptera as preys. On the other hand, this could mean that Lepidoptera scales were ON the insects and not inside. Furthermore, the report of parasitoid Hymenoptera could mean that the predator insects have been parasitised by the wasps as opposed to these being their food. I have no information on the ecology of the wasps recorded by the authors, but they should provide information on why they assume this is food and not result of a parasitised predator.

The second main issue with this work is the metabarcoding analysis.

In particular, the number of reads reported for the vast majority of 'recorded insect preys' is too low. In many cases, the number is as low as just 2 reads: not enough to draw the conclusion this can be surely a positive result. I understand this is strongly due to the fact the vast majority of reads is produced by the predator itself, but unfortunately a number of reads ranging between 2 and 30/50 cannot be considered a true positive. For example, this could be a clear case of index switching during the metabarcoding run. Unfortunately, the authors did not analyse replicates from each sample, which could have provided more certainty for low numbers. 

For example, the fact that of the 79 samples of Forficula senegalensis analysed, only a single one reported 2 reads of C. pallidus and only a single other sample reported just 3 reads of C. barbarus suggests these reads are the result of contamination.

Ultimately, these two major issues make the paper in the present form not fit for publication.

However, my suggestion to the authors could be to separate the data of the bats/birds from the predator insects. The DNA extraction from the feces seems more robust (despite this not being my field). Hence, with a stricter data analysis (e.g. cut off threshold of at least 100 reads?) the results could be robust enough for publication and the work equally interesting.

I would strongly encourage you not to give up and keep up the great ideas that were at the base of this work!

Author Response

Concerning your first concerns

We did not dissect the insects for four reasons, (1) dissection is very laborious for a relative gain and (2) it can be a major source of contamination (see King et al., 2008; Pons, 2006). In addition, potential co-amplifications may exist in any cases on the cuticle of ingested prey. As a result, in this type of study, sources of contaminations with traces of DNA of non-direct prey are expected at least in small proportion.

Measures have been taken to avoid potential cross-contamination, i.e. individualized sampling and storage of insect samples, placing the samples at -20°C first before putting them in ethanol to avoid regurgitation, which I consider to be the most important source of contamination (line 83-84).

We added this paragraph to the discussion: Line 300-304. “In this study, we assume that the use of whole body of arthropod samples may involve possible contamination from the exoskeleton (i.e. DNA outside the body) of these arthropods. Nevertheless, precautions such as systematically individualizing samples during field sampling and placing live samples at -20°C before storage in alcohol to avoid regurgitation were taken to avoid potential external contamination of arthropod samples [42]”.

The term ‘gut content’ were deleted to the text for arthropod predators, and was used to refer faeces of birds and bats , see line 108-109: “In this paper, the term 'gut content' will be used to refer the faeces of birds and bats”

Concerning your second concerns

In our study, all samples were analyzed in triplicate and therefore the taxa conserved are those only observed in at least 2 independent replicates out of 3 (see line 173-176) (Robasky et al., 2014). In addition, a series of non-arbitrary filters were applied following the recommendation of (Galan et al., 2018). This approach is much more robust to effectively eliminate false positives (see (Alberdi et al., 2018; Corse et al., 2017), as each OTU is filtered according to its abundance in the sequencing run and better sensitivity is preserved for rare or low biomass prey.

We don't know the half-life time of the prey DNA in the predator's gut, but if it's short we have little chance to detect something. Capturing an arthropod predator just after consuming prey is quite improbable, which probably explains the low rate of individuals giving a result.

Reviewer 2 Report

The paper describes the dietary analysis of arthropod and vertebrate predators enabling the identification of predators involved in the control of arthropod pests. The paper is well written with only minor corrections in the language. This is a timely study and I recommend the publication of this study. I don't have any major criticism of this study. 

Author Response

References have been corrected

A proofreading of English by a 'native' has been done

Reviewer 3 Report

The manuscript is well written, with clear aims and adequate methodological framework. Results are appropriately analyzed and discussion is well presented. Addressing the system such as millet fields is of interest as this crop is of economic importance on larger scales. This manuscript provides novel data on this cropping system, but also clearly emphasizes the usefulness of this methodological approach in delineation of multi-trophic interactions in ecosystems. I find it acceptable for publication in original form.   

Author Response

A proofreading of English by a 'native' has been done

Round 2

Reviewer 1 Report

I appreciate the efforts the authors put in this reviewed version of their manuscript. In particular I appreciate the thorough explanation of the number and type of controls used in their metabarcoding analysis. Unfortunately, this is not enough to eliminate my concerns on the present manuscript.

If the paper aims to disentangle food webs, then it is assuming that the DNA reads coming from the insects are from their food, hence from the insects’ gut. Unfortunately, the authors cannot prove this.  The fact that the authors agreed on deleting the term ‘gut content’ means that they (I suppose) agree this term cannot be used in this study to describe the reads they obtained from the insects. If the reads are not from the gut content, then they are not from something the insect ate. Consequently, this is not what the aim of the paper was studying.

Additionally, answering the authors’ reply concerning my first issue (the contamination):

-the authors list papers from 2006 and 2008 as reference suggesting that dissection is source of contamination. These references are dated pre-metabarcoding technology, so they really have scarce value in this context. A dissection operated for each single insect in a contamination-free area is possible (i.e., under a hood cleaned with bleach).

-The authors suggest that dissection is laborious, which is surely true, but they also say that it brings only a relative gain. I do not agree with this. Separating the gut from the rest of the body would have contributed to less host’s DNA being amplified and, if the reads they obtained were actually from the gut, these would have been in a higher proportion (which then links to my second doubt, the number of reads for some of this is too low).

-The added paragraph is unfortunately not enough. The authors admit that contamination from exoskeleton is possible. Consequently, when their results are based on 1 to 5 reads, this is either a contamination OR an actual prey. It cannot be both. Since they obviously cannot say which is which, due to the very low number of reads, then the discussion based on the result is flawed.

Let’s take the examples of the reads belonging to Creontides pallidus. The authors recorded 2147 reads in one of their samples of Orius sp. and more than 300 in another. This samples have enough reads to be true positive. However, only 2 reads were recorded in a single specimen of Forficula sp. I would hypothesize this might be the result of index switching. A higher threshold should be applied for the number of reads that are considered a certain true positive result. Surely, we cannot put on these 2 reads the same meaning we confidently put on more than 2000.

In my opinion, the authors still have the extremely valid option of removing the part of the insect predators and keep the part on birds and bats. I think this is a very valuable content and, being based on the droppings of these animals, they can be sure of their results.

Unfortunately, I have to suggest this paper is not published in its present form.

Author Response

The samples were not rinsed, but when transferring the samples to the extraction plates, care was taken to wipe them on cotton previously soaked in 20% diluted bleach. This has most likely also eliminated contamination on the exoskeleton of arthropods, which justifies that all DNA detected after filtering and triplicate (even those with a low number of readings) is necessarily found in the stomachs of insects (prey, secondary prey or parasitoids).
We have therefore added this paragraph to the methodology section ‘The samples were wiped with cotton wool previously soaked in 10% bleach before extraction’ (lines 98-99).

We have decided to delete the term "gut content" for arthropods to make it easier for the reader to understand, who will ask the same questions as you. However, we do not doubt the accuracy of our results. We confirm that there are no problems with index changes, but real positive points, as all occurrences are validated by at least 2 independent sequences out of 3 using technical replicas.
In addition, the same critiques can also be made on fecal pellet samples (which, however, you have no problem with). Environmental contamination by "external" insects is possible with insects (e.g. coprophages or their feces) present on the samples at the time of sampling (see Galan et al. 2018). There is no reason why this should be more important for predatory insects than for feces collected from the soil.